# qPCR Detection and Quantification of *Aggregatibacter actinomycetemcomitans* and Other Periodontal Pathogens in Saliva and Gingival Crevicular Fluid among Periodontitis Patients

**DOI:** 10.3390/pathogens12010076

**Published:** 2023-01-03

**Authors:** Sarah Reddahi, Amal Bouziane, Kaoutar Dib, Houssain Tligui, Oum keltoum Ennibi

**Affiliations:** 1Research Laboratory in Oral Biology and Biotechnology, Faculty of Dental Medicine, Mohammed V University in Rabat, Rabat 10000, Morocco; 2Laboratory of Biostatistics, Clinical Research and Epidemiology, Department of Periodontology, Faculty of Dental Medicine, Mohammed V University in Rabat, Rabat 10000, Morocco; 3Medical Research Laboratory, Children’s Hospital, University Centre Ibn Sina, Faculty of Medicine and Pharmacy, Mohammed V University in Rabat, Rabat 10000, Morocco; 4Research Laboratory in Oral Biology and Biotechnology, Department of Periodontology, Faculty of Dental Medicine, Mohammed V University in Rabat, Rabat 10000, Morocco

**Keywords:** subgingival plaque, periodontitis, qPCR, bacteria, saliva, *A. actinomycetemcomitans*, *JP2* clone, *P. gingivalis*

## Abstract

Objective: The detection of special bacterial species in patients with periodontitis is considered useful for clinical diagnosis and treatment. The aim of this study was to investigate the presence of specific periopathogens and investigate whether there is a correlation between the results of different bacterial species in whole saliva and pooled subgingival plaque samples (healthy and diseased sites) from individuals with periodontitis and periodontally healthy subjects. Materials and methods: In total, 52 patients were recruited and divided into two groups: non-periodontitis and periodontitis patients. For each group, the following periodontal pathogens were detected using real-time polymerase chain reaction: *A. actinomycetemcomitans JP2* clone, *A. actinomycetemcomitans non JP2* clone, *Porphyromonasgingivalis*, and total eubacteria. Results: Higher levels of the various studied bacteria were present in both saliva and plaque samples from the periodontitis group in comparison to non-periodontitis subjects. There were significant differences in *P. gingivalis* and *A. actinomycetemcomitans JP2* clones in the saliva of periodontitis patient compared to the control group. Subgingival plaque of diseased sites presented a significant and strong positive correlation between *A. actinomycetemcomitans* and *P. gingivalis*. In saliva samples, there was a significant positive correlation between *A. actinomycetemcomitans JP2* clone and *P. gingivalis* (*p ≤* 0.002). Conclusion: Quantifying and differentiating these periodontal species from subgingival plaque and saliva samples showed a good potential as diagnostic markers for periodontal disease. Regarding the prevalence of the studied bacteria, specifically *A. actinomycetemcomitans JP2* clone, found in this work, and the high rate of susceptibility to periodontal species in Africa, future larger studies are recommended.

## 1. Introduction 

Periodontitis is a chronic inflammatory condition that occurs in the presence of multiple etiological factors, including mainly bacteria and immune response; but also, genetic, and environmental factors [1]. In this complex scheme, the presence of specific bacteria in the subgingival biofilm may play a key role in the pathogenesis of the disease, either directly or indirectly. The subgingival biofilm is a complex microbiota with over 700 bacterial species [2]. Among these, a distinct microbial community dominated by obligate anaerobes and proteolytic species, namely *Porphyromonas gingivalis* [3], *Treponema denticola* and *Tannerella forsythia*, known as the “red complex”, and *Aggregatibacter actinomycetemcomitans*, have been shown to be risk factors for the onset or progression of periodontitis [4,5,6,7,8,9,10]. Furthermore, Oliveira et al. and Chapparo et al. [11,12], added 13 bacterial taxa to the list of potential periodontal pathobionts based on the results of open-ended 16S rRNA gene analyses. The presence of the putative bacteria in gingival crevicular fluid (GCF) has been linked to diseased sites and in low numbers to healthy sites. Likewise, high concentrations of this bacteria are found in saliva of periodontitis patients [13]. 

Thus, sampling the saliva might be a promising method of microbial diagnosis in periodontal conditions [14,15], as this fluid can be easily, repetitively, and non-invasively collected. However, several studies have shown that the bacterial composition of saliva differs significantly from that of dental biofilm, including both supra- and subgingival microbiota [16,17]. A previous study also revealed a difference in microbiota community structure between supragingival plaque and saliva, as well as the compositional stability of salivary microbiota against a supragingival microbiota shift [18]. 

The aim of this case control study was to assess the presence and quantification of *A. actinomycetemcomitans clone JP2* and *non JP2*, *P. gingivalis* and total bacteria in saliva and gingival crevicular fluid in periodontitis patients.

## 2. Material and Methods

### 2.1. Study Population and Clinical Measurements

This case control study included 32 patients seeking periodontal treatment at the clinical department of periodontology in the Center of Consultation and Dental Treatments (CCTD), Ibn Sina university hospital in Rabat. Twenty-one non-periodontitis subjects were also recruited as a control group. Inclusion criteria were, systemically healthy subjects, aged 18 years old and above, non-tobaccousers, and with at least 20 teeth. Exclusion criteria were use of antibiotics and or periodontal treatment during the past six months prior to the clinical examination, and pregnant or lactating women. 

The Ethical Committee for Biomedical Research at Mohammed V University Faculty of Medicine and Pharmacy in Rabat approved this study protocol(N°62/18) in accordance with the Helsinki Declaration on research involving human subjects, and all participants signed an informed consent form written in their native language.

### 2.2. Study Design

For periodontal diagnostics, probing depth (PD) and clinical attachment loss (CAL), were measured at six sites per tooth, excluding third molars using a standard periodontal probe (Hu-friedy, Chicago, IL, USA). Plaque index (PI) (O’Leary et al. 1972) [19] and bleeding on probing (BOP) (Ainamoand bay 1975) [20] were assessed, too. Retro-alveolar X-rays were taken for periodontitis patients.

The absence of periodontitis was defined as absence of history periodontitis, PD ≤ 3 mm, no proximal clinical attachment loss, and no recession. Whereas a periodontitis patient was defined as having at least two non-adjacent teeth with interdental clinical attachment loss (CAL) ≥2 mm, probing pocket depth (PPD) of more than 3 mm, and radiographic evidence of bone loss [21,22]. 

Periodontitis were classified by stage and grade accordingly to the latest classification of periodontal diseases and conditions [22]. Definitions of stages were based on severity (primarily periodontal breakdown and periodontitis-associated tooth loss), complexity of management (pocket depth, infrabony defects, tooth mobility, furcation defects, masticatory deficiency), when grades definitions of periodontitis were based on direct or indirect evidence of the progression rate of the disease. Three categories can be distinguished: slow (Grade A), moderate (Grade B), and a rapid rate of progression (Grade C) [22].

### 2.3. Saliva and Subgingival Plaque Sampling 

Saliva and gingival crevicular fluid (GCF) were collected from each participant.

For saliva sampling, all subjects were instructed to avoid using oral hygiene products and to abstain from eating for at least one hour before the examination. Salivary sampling was conducted from 08:30 a.m. to 11:30 a.m. After rinsing the oral cavity with tap water for 30 s and passive drooling into sterile plastic tubes, 5 mL of unstimulated whole saliva was collected from each subject. Immediately, collected saliva samples were centrifuged for 15 min at 6000 rpm, then distributed in aliquots and frozen at −80 °C until analysis [23,24]. 

GCF samples were collected as pool samples. In the periodontitis group, we collected one pooled sample from diseased sites (two deepest pockets exhibiting the highest CAL values) and one pool sample from two healthy sites when possible. A pool sample was taken from two healthy sites in healthy subjects too. Sampling sites were isolated from the saliva by a cotton roll and supra-gingival plaque was removed, after that two sterilized autoclaved medium paper point N°40 were inserted consecutively into the chosen sites and left for 30 s. Samples were then put in 500 µL of phosphate-buffered saline (PBS) solution [24]. 

All samples were centrifuged for 15 min at 6000 rpm before being frozen at −80 °C until microbial analysis. 

### 2.4. Microbiological Analysis

Both saliva samples and subgingival plaque samples were analyzed by qPCR. Specific primers were used for the identification and quantification of the following periodontal pathogens: *Porphyromonasgingivalis (Pg)*, *A. actinomycetemcomitans JP2 (Aa JP2)*, and *A. actinomycetemcomitans non JP2 (Aa non JP2).* A universal probe detecting 14 oral bacterial species (*A. actinomycetemcomitans*, *Porphyromonasgingivalis*, *Treponema denticola*, *Tanerella forsythia*, *Fusobacterium nucleatum*, *Prevotella intermedia*, *Haemophilusaphrophilus*, *Eikenellacorrendens*, *Streptococcus (S.) anginosus*, *S. sobrinus*, *S. gordonii*, *S. mutans*, *S. salivarius*, *Escherchia coli.*) were also used.

*P. gingivalis (Pg)* (clinical purified strain), *A. actinomycetemcomitans JP2 (Aa JP2)* (HK 912), and *A. actinomycetemcomitans non JP2 (HK 1605*) were used as positive control strains. They were obtained from Umeå University’s Department of Odontology in Sweden.

Standard suspensions ranging 10^8^ to 10^1^ cells/mL were prepared for *P. gingivalis*, *A. actinomycetemcomitans JP2*, and *A. actinomycetemcomitans non JP2*.

Bacterial DNA was extracted from straincultures, saliva and subgingival plaque samples, using the Invitrogen PureLink Genomic DNA Mini Kit (Thermo Fisher Scientific, Waltham, MA, USA), as directed by the manufacturer. A spectrophotometer was used to determine the purity (quality) and yield (quantity) of the eluted DNA (Nanodrop, Thermo Fisher Scientific, Waltham, MA, USA). The absorbance ratio at 260 and 280 nm (A260/A280) was used to determine the purity of DNA.

### 2.5. Primers and Probes 

The selection of primers was based on previous data sources with information on species-specific oligonucleotides of the 16S ribosomal RNA in each strain of this study [25,26]. Table 1 shows the primer pairs and TaqMan probe sets used in the real-time PCR assay for the detection and quantification of the targeted periodontal pathogens. The expected product size is shown in Table 1. The primers were purchased from Invitrogen by Thermo Fisher Scientific. 

### 2.6. Real-Time PCR Assay

Quantitative real-time PCR assay was performed using the Applied Biosystems Quant Studio 5 Real-Time PCR System with the cycle profile shown in Table 2. Each PCR was performed in a total volume of 20 μL of PCR mixture using 10 μL qPCR kits (2×TaqMan Multiplex Master Mix, Applied Biosystems, Foster City, CA, USA), 0.8 μL each of sense and antisense primers, 1 μL of the probe, 3.4 μL of sterilized DNase-RNase-free water, and 4 μL of template DNA solution. The NTC includes all of the RT-PCR reagents except the DNA template.

Quantification analysis uses sample “crossing points” (CP) to determine the presence and the concentration of the target DNA in known and unknown samples after amplification. Threshold cycle (Ct) is defined as the number of cycles required for the fluorescence to become detectable above the background fluorescence and Ct levels are inversely proportional to the logarithm of the amount of target nucleic acid in the sample. Standard curves were generated for all the target bacteria using DNA from pure cultures and species-specific DNA primers. DNA isolated from a pure culture of *P. gingivalis* was used in generating a standard curve for the universal primers and probe (Appendix A). Bacterial DNA levels were calculated from the standard curve equations created by tenfold serial dilution of DNA standards. All samples, including DNA standards, subgingival plaque and saliva samples, and negative test control were analyzed in duplicate.

### 2.7. Data Analysis

Statistical analysis was performed using Jamovi software [27]. Values of demographic features and clinical parameters are presented as the mean ± standard deviation (SD) or percentage.

We used Student t-tests for the comparison of clinical parameters (BOP, pockets depths, Clinical attachment loss) between control and patient groups.

The Mann–Whitney test was used to compare the concentrations of studied bacteria in healthy subgingival plaque samples and saliva between the non-periodontitis and periodontitis groups.

Khi-deux and Fisher tests were used to determine the frequency of qualitative variables in the two group populations.

Pearson’s correlation test was used for quantitative variables with a Gaussian distribution, and for variables with a non-Gaussian distribution, we used Spearman’s correlation.

A *p* value of less than 0.05 was considered statistically significant.

## 3. Results

The mean age of the study population was 27.83 ± 8.18, among them 77.4% were female. Table 3 summarizes the socio-demographic and clinical characteristic of the studied groups. 

In periodontitis group, five patients had periodontitis Grade B and 27 patients had periodontitis Grade C.

Global results regarding the prevalence of the studied bacteria showed an absence of *A. actinomycetemcomitansnon JP2* and *Aa JP2* clones in non-periodontitis patients both in saliva and sub-gingival plaque. In periodontitis patients, *Aa non JP2* was present in saliva and in healthy sites (respectively 16% and 10%), while *Aa JP2* was present in 50% of saliva samples and 16.7% in healthy sites.

*Pg* was present either in non-periodontitis patients and periodontitis patients both in saliva and healthy sites. The prevalence of *Pg* was significantly higher in saliva of periodontitis patients (70.1%) versus 38.1% in non-periodontitis patients (*p* = 0.024). In healthy sites, there was no statistical difference regarding the presence of *Pg* between the two groups (50% for non-periodontitis patients versus 56.7% in periodontitis patients).

The level comparison of the different quantified bacteria in saliva and healthy sites of the tow studied groups showed mainly: 

A high concentration of eubacteria in periodontitis patients’ healthy sites compared to the control group (*p <* 0.001).

There was a statistically significant difference in *P. gingivalis* and Aa *JP2*clone concentrations in saliva samples, with the diseased group having a higher concentration than the non-periodontitis group (Table 4). 

In the periodontitis group, the results showed a statistically significant difference in all quantified bacteria between healthy sites and diseased sites (*p <* 0.005) (Table 5). 

When comparing Grade B and Grade C periodontitis, despite being statistically insignificant, the results showed a high concentration of total bacteria, *Pg*, *Aa JP2* and *non-JP2* clones in the patient group with grade C periodontitis compared to the grade B group (Table 6).

The correlations between the different bacterial species in all individuals are shown in Table 7. The correlation coefficients ranged from −1 to 1, with positive correlations being greater than 0 and negative correlations being lower than 0. In the subgingival fluid of diseased sites, *P. gingivalis* demonstrated a substantial strong positive correlation (rs = 0.934) with *Aa non JP2* bacteria (*p* < 0.001). The correlation between *Aa non JP2* and *P. gingivalis* in the subgingival fluid of healthy sites was (Spearman r = 0.377), with a *p* value of 0.007, indicating a significant medium-positive correlation. 

In saliva samples of the whole population, *JP2* and *P. gingivalis* bacteria showed a significant positive correlation (rs = 0.432; *p =* 0.002) and a very significant (*p* < 0.001) positive correlation between *JP2* and *Aa non JP2* (rs = 0.505) (Table 8).

## 4. Discussion

Higher levels of the studied periopathogens were found in both saliva and plaque samples from the periodontitis group in comparison to healthy subjects. This supports previous studies on the bacterial etiology of periodontitis. Indeed, it is largely admitted that periodontitis may occur in presence of complex polymicrobial communities, which can threaten the periodontium because the grow of the total number of bacteria or development of highly pathogenic bacteria. The complex polymicrobial biofilm associated to periodontitis interacts with the immune response, leading to periodontium breakdown.In the present study, we noticed a significant difference in *P. gingivalis* and *A. actinomycetemcomitans JP2* clone in the saliva of the periodontitis patients compared to the control group, and statistically significant differences were also found for all the studied bacteria in subgingival plaque samples of periodontitis when compared to non-periodontitis patients. 

In the gingival fluid of diseased sites, there was a significant and strong positive correlation between *A. actinomycetemcomitans non JP2* and *P. gingivalis*. The same bacteria were found in the gingival fluid of our population’s healthy sites and showed a positive correlation. *JP2* clone and *P. gingivalis* bacteria showed a significant positive correlation in saliva samples, and *JP2* and *Aa* bacteria showed a strong correlation.

Microbiological identification by colony counting has long been regarded as the “gold standard” technique. However, this is clearly limited due to the complex nature of periodontal disease bacterial flora, which makes it extremely difficult to cultivate [28,29]. qPCR, on the other hand, is a more sensitive and accurate method for detecting and quantifying periodontal pathogens in various oral samples. Much of the previous research has focused on investigating subgingival microbiota levels using a case–control study design. The control group may include healthy subjects [30], gingivitis patients [31], or only different periodontal patient groups based on periodontal severity [32]. Furthermore, some qPCR studies have been conducted solely on subjects with periodontal disease [33]. Thus, traditional methods of diagnosing periodontal disease are insufficient, with recently more detailed stages and grades corresponding to the associated treatment protocol, a microbiological examination that is characteristic of the entire oral cavity’s subgingival microflora is important for the use of concomitant systemic antibiotics in certain types of periodontitis [34,35]. 

Due to its easy, quick, and non-invasive sampling, salivary microbiota quantification has been successfully used in several studies and has been proposed as a diagnostic biomarker for periodontitis [36,37] either in the healthy population [38] or in the periodontitis patients [39]. However, the number of studies that have used the conventional quantitative real-time PCR (qPCR) technique to detect and quantify oral pathogens in distinct study groups is limited. 

Our results are consistent with previous research that found a higher number of pathogens in periodontal patients than in healthy patients [37,40,41]. Many studies have heterogeneous bacteria analysis results in both types of sampling because different techniques, study population types, and saliva sampling methods were used. They revealed differences in the mean prevalence of subgingival or saliva species in samples from subjects with various clinical conditions, which could be due to differences in geographic locations, diet, genetics, and disease susceptibility factors [42,43]. However, and according to the available literature, there is a notable underrepresentation of data from the Middle East and North Africa geographic region (MENA) [44]. Specifically, no observational studies on adults or evaluating the prevalence and number of salivary periodontal microorganisms in African countries, particularly the Moroccan population, have been conducted.

Consequently, in this study, we investigated the presence and the amounts of three important periodontal pathogens, namely *A. actinomycetmcomitans non JP2*, *P. gingivalis*, and *A. actinomycetmcomitans JP2* clones, as well as the total number of bacteria estimated with universal primers and probes in subgingival plaque and whole saliva samples from periodontitis and non-periodontitis subjects. A correlation analysis among the aforementioned microorganisms was also confirmed. 

*Porphyromonas gingivalis* (*P. gingivalis*) has been reported to be mainly prevalent in Asian populations [45]. As a member of the red-complex bacteria, it has been the most frequently studied species, along with *Tannerella forsythia* and *T. denticola*, and has been strongly associated to periodontitis. Most investigations reported more than 60% of red-complex species in saliva and subgingival plaque samples from periodontitis patients [41,46]. Our findings are comparable, with more than 70% prevalence in saliva and plaque samples of the patient group and a 100% detection rate in subgingival sites with periodontal damage. Regardless of sample type, the detection frequencies of *P. gingivalis* in the healthy group were much lower than those in the corresponding periodontitis group.

Generally, most studies reported that the detection frequency of specific red-complex bacteria in periodontal disease patients was higher in subgingival plaque than in saliva, which is in accordance with our results. However, several studiesfound the opposite shift [47,48]. 

Almost all correlations between bacteria and with the PPD and CAL values were positive. It explains that when a subgingival site with periodontal destruction was present, the majority of putative periopathogens presented higher values, as shown in Table 9. Comparison of the relative abundance in plaque samples showed a significant medium positive correlation between *P. gingivalis* and *A. actinomycetemcomitans* in the healthy sites while a substantial strong positive association was observed between *P. gingivalis* and *Aa* (rs = 0.934, *p* < 0.001). Analysis of salivary samples, on the other hand, revealed a very strong correlation between *JP2* clone and *Aa* bacteria, as well as *P. gingivalis* and *JP2* clone, respectively (*p* < 0.001, *p* = 0.002). In each correlation, both bacterial species have an effect on each other’s growth. 

Our results are in line with several previous cross-sectional studies [13,48,49]. Sereti et al. [50] found a positive correlation on the level of the six microorganisms studied (i.e., *A. actinomycetemcomitans* (*Aa*), *P. gingivalis* (*Pg*), *Tannerella forsythia* (Tf), *Treponema denticola* (*Td*), *Parvimonasmicra* (*Pm*), and *Prevotella intermedia* (*Pi*)) in plaque and saliva samples of periodontitis patient groups except for *Aa* which was not detected. Based on a longitudinal study, Haubek and al. 2008 showed that subjects who carried the *JP2* clone of *A actinomycetemcomitans* alone or together with *non-JP2* clones of *A actinomycetemcomitans* had a significantly increased risk of periodontitis. Meanwhile, the risk of developing the disease was lower in those carrying *non-JP2* clones only [9]. Moreover, Belibasakis et al. 2019 reported that when integrated within a 10-species subgingival biofilm model, *A. actinomycetemcomitans* did not significantly impact the abundance of the other bacterial species, nor did it affect the biofilm structure [51].

*A. actinomycetemcomitans* (*Aa*) is known as one of the most putative periodontal pathogens [52]. However, its prevalence varies greatly between studied populations worldwide. Disparities between countries may be due to differences in diet, periodontal status of the carriers, genetics, and ethnic racial characteristics of the population. In this study, the periodontitis group harbored higher prevalence of *Aa non JP2* and *JP2* clone both in saliva and diseased sites, whereas they were completely absent at the healthy sites as well as in samples from non-periodontitis subjects. These findings are consistent with those of other studies [53]. Some studies showed a low *Aa* detection frequency [54,55,56]. 

*A. actinomycetemcomitans* is an oral bacterium of the green complex that is involved in the pathogenesis of aggressive periodontitis in younger patients [43,56,57,58]. The difference in detection prevalence of this pathogen, despite being strongly associated with periodontal disease development and progression, may be due to *A. actinomycetemcomitans*’ serological heterogeneity. *Aa* serotype b, also known as the *JP2* genotype, is the most pathogenic of the seven publicly known serotypes due to its extremely high levels of leukotoxin production, the main virulence factor of *Aa* when compared to the other serotype species [56,57,58]. 

In a recent cross-sectional study, Thorbert-Mrosetet al. used qPCR to assess the prevalence and quantity of *A. actinomycetemcomitans* (and the *JP2* clone in particular) and five other bacterial subgingival species in children and adolescents from two groups of participants: Somali subjects and Swedish subjects of non-Somali ancestry. *A. actinomycetemcomitans* was present in almost all Somali participants. Furthermore, the *JP2* clone was discovered in five Somalis (including two periodontitis cases), confirming the clone’s association with African populations. *P. gingivalis* and *Treponema denticola* were found in significantly higher numbers and frequencies in the Somali group, indicating a mature and adult type of subgingival microbiota [59]. Our results as well as previous research backed up these findings, demonstrating that *JP2* genotype carriers may be exclusively found in people of African descent [58,60,61,62]. 

To our knowledge, this is the first study to successfully demonstrate the presence and prevalence of *P. gingivalis*, *Aa non JP2*, *Aa JP2* genotype, and eubacteria species in a case control Moroccan community. It is also the first study to look at these putative pathogens in saliva and subgingival healthy and diseased sites from the same population. Our findings showed that the real-time PCR protocol described here was effective for quantifying and differentiating the periodontal species from subgingival plaque and saliva samples. 

We did not evaluate all periopathogenic bacteria in this study, and given the high rate of susceptibility to periodontal pathogens in the African and MENA regions, additional studies involving more periodontal species using a larger population range would be useful for understanding the complex ecology observed in periodontitis and could also be used as biomarkers for the diagnosis and treatment of this disease.

## Figures and Tables

**Table 1 pathogens-12-00076-t001:** Primers/TaqMan probes for real-time PCR.

Primers/Probe Sets	Product Size (bp)	Target	References
Universal 5′-GATTAGATACCCTGGTAGTCCAC-3′ 5′-TACCTTGTTACGACTT-3′ 5′-FAM-CACGGTGAATACGTTCCCGGGC-TAMRA-3′	69	16SrDNA	[25]
*A. actinomycetemcomitans non JP2* 5′-CGCAAGTGCCATAGTTATCCACT-3′ 5′-TCGTCTGCGTAATAAGCAAGAGAG-3′ 5′-FAM-ATATTGTAGACATCGCCC-MGB-3′	145	16SrDNA	[26]
*A. actinomycetemcomitansJP2* 5′-TCT ATG AAT ACT GGA AAC TTG TTC AGA AT-3′ 5′-GAA TAA GAT AAC CAA ACC ACA ATA TCC-3′ 5′-FAM-ACA AAT CGT TGG CAT TCT CGG CGA A-TAMRA-3′	151	ltxA	[26]
*P. gingivalis* 5′- AGGCAGCTTGCCATACTGCG- 3′ 5′- ACTGTTAGCAACTACCGATGT-3′ 5′-FAM GCTAATGGGACGCATGCCTATCTTACAGCT-TAMRA-3′	404	16SrDNA	[25]

FAM 6-carboxyfluorescein, TAMRA tetramethylrhodamine, MGB minor groove binding.

**Table 2 pathogens-12-00076-t002:** qPCR cycle program settings.

Bacteria	Cycles Settings	
Universal	Hold/time: Cycling/time: Cycling/time: Cycling/time: Cycles:	95°/10 min 95°/15 s 54° 30 s 72° 30 s 45
*A. actinomycetemcomitans* *non JP2*/*JP2*genotype	Hold/time: Cycling/time: Cycling/time: Cycling/time: Cycles:	95°/10 min 95°/15 s 58°/40 s 72°/30 s 40
*P. gingivalis*	Hold/time: Cycling/time: Cycling/time: Cycling/time: Cycles:	95°/10 min 95°/15 s 58°/30 s 72°/30 s 40

**Table 3 pathogens-12-00076-t003:** Sociodemographic and clinical characteristics of the population.

Variables	Non-Periodontitis Patients (21 (23.6))	Periodontitis Patients (32 (60.4))	*p* Value
Age (years) *	24.67 ± 1.82	29.91 ± 9.94	0.021
Gender ** Male Female	2 (9.5) 19 (90.5)	10 (31.3) 22 (68.8)	0.095
BOP *	29.57 ± 10.20	93.71 ± 12.88	<0.001
Total mean of Probing *	2.35± 0.25	5.54 ± 0.83	<0.001
Total mean of CAL *	----	3.85 ± 1.39	*-----*
PD of healthy sampling sites (mm) *	1.90 ± 0.30	2.14 ± 0.48	0.061
PD of diseased sampling sites (mm) *	----	7.07 ± 1.36	*------*
CAL in sampling sites (mm) *	------	6.35 ± 2.26	*----*

*: Mean ± SD; **: *n* (%); PD: probing depth; CAL: clinical attachment level.

**Table 4 pathogens-12-00076-t004:** Quantification levels of studied bacteria in saliva and healthy sites in non-periodontitis versus periodontitis subjects.

Bacteria	Non-Periodontitis (*n* = 21)	Periodontitis (*n* = 32)	*p* Value
Universal * Healthy Sites (copies/µL) Saliva (copies/µL)	16.8 × 10^6^ [14.0 × 10^6^ − 27.0 × 10^6^] 660,681 [516,038–728,127]	35.6 × 10^6^ [24.8 × 10^6^ − 43.3 × 10^6^] 665,965 [582,570–738,491]	<0.001 1.00
*A. actinomycetemcomitans* *non JP2* *			
Healthy sites (copies/µL)	0.00 [0.00–0.00]	0.00 [0.00–0.00]	0.052
Saliva (copies/µL)	0.00 [0.00–0.00]	0.00 [0.00–0.00]	0.139
*P. gingivalis* * Healthy sites (copies/µL) Saliva (copies/µL)	19.80 [0.00–1167] 8.00 [0.00–103]	179 [0.00–283,121] 236 [2.00–10,122]	<0.001 <0.001
*A. actinomycetemcomitans JP2* clone * Healthy sites (copies/µL)	Absent	Absent	NA
Saliva (copies/µL)	0.00 [0.00–0.00]	1.50 [0.00–116]	<0.001

*: Median and interquartile range [IQR]; NA: not applicable.

**Table 5 pathogens-12-00076-t005:** Comparison of the number of bacteria in the subgingival plaque between diseased and healthy sites in periodontitis patients.

Bacteria (Copies/µL)	Healthy Sites (*n* = 32)	Diseased Sites (*n* = 32)	*p* Value
*Pg* *	179 [0.00–283,121]	1.52 × 10^7^ [100,657–4.43 × 10^7^]	0.003
Universal *	3.56 × 10^7^ [2.48 × 10^7^–4.33 × 10^7^]	5.04 × 10^7^ [4.66 × 10^7^–5.46 × 10^7^]	<0.001
*Aa JP2* *	0.00 [0.00–0.00]	2.00 [0.00–413]	<0.001
*Aa non JP2* *	0.00 [0.00–0.00]	5.00 [0.00–2.78 × 10^6^]	0.001

*: Median and interquartile range [IQR]; *Aa*: *Aggregatibacter actinomycetemcomitans*; *Pg*: *Porphyromonas gingivalis*.

**Table 6 pathogens-12-00076-t006:** Bacteria count in subgingival plaque samples from diseased sites in patients with Grade B versus Grade C periodontitis.

Bacteria (Copies/µL)	Grade B Patients (*n* = 5)	Grade C Patients (*n* = 27)	*p* Value
*Pg* *	1.54 × 10^7^ [254.30–4.32 × 10^7^]	1.50 × 10^7^ [115,216.08–4.55 × 10^7^]	0.420
Universal **	4.70 × 10^7^ ± 3.05 × 10^6^	5.19 × 10^7^ ± 5.70 ×10^6^	0.075
*Aa JP2* *	2.00 [0.50–1.8 × 10^4^]	1.50 [0.00–3.14 × 10^6^]	0.905
*Aa non JP2* *	116.97 [0.00–1.88 × 10^6^]	2.50 [0.00–3.97 × 10^6^]	0.854

*: Median and interquartile range [IQR]; **: Mean ± SD; *Aa*: *Aggregatibacter actinomycetemcomitans*; *Pg*: *Porphyromonas gingivalis*.

**Table 7 pathogens-12-00076-t007:** Interspecies correlations in subgingival plaque of all subjects.

	*Pg* DS	*Pg* HS	Univ. DS	Univ. HS	*Aa JP2* DS	*Aa JP2* HS	*Aa non JP2* DS	*Aa non JP2* HS
*Pg* DS			0.105	−0.064	0.001	NA	0.934 *	0.255
*Pg* HS			−0.046	−0.022	−0.041	NA	0.266	0.377 **
Univ. DS					0.165	NA	0.066	0.051
Univ. HS					0.007	NA	−0.097	0.247
*Aa JP2* DS							0.049	−0.102
*Aa JP2* HS							NA	NA
*Aa non JP2*DS								
*Aa non JP2HS*								

*Aa*: *Aggregatibacter actinomycetemcomitans*; *Pg*: *Porphyromonas gingivalis*; Univ: Universal; DS: diseased sites; HS: healthy sites. * *p* < 0.001; ** *p* = 0.007.

**Table 8 pathogens-12-00076-t008:** Interspecies correlations in the saliva of all subjects.

	*Pg*	Universal	*Aa JP2*	*Aa non JP2*
*P. gingivalis*		0.184	0.432 *	0.261
Universal			0.202	0.199
*Aa JP2*				0.505 **
*Aa non JP2*				

*Aa*: *Aggregatibacter actinomycetemcomitans*; *Pg*: *Porphyromonas gingivalis*; * *p* = 0.002; ** *p* < 0.001.

**Table 9 pathogens-12-00076-t009:** Sociodemographic and clinical characteristics of periodontitis Grade B and Grade C patients.

Periodontitis Patients (*n* = 32)	Grade B 5 (15.6)	GradeC 27 (84.4)	*p* Value
Age (years) *	33.60 ± 10.40	29.22 ± 9.97	0.375
Gender ** Male Female	30.0% 9.1%	70.0% 90.9%	0.293
Mean of PD	4.88 ± 0.26	5.64 ± 0.84	0.089
Mean of CAL	2.79 ± 0.80	4.01 ± 1.40	0.103
PD of sampling sites *	6.25 ± 0.50	7.19± 1.41	0.202
CAL of sampling sites *	4.50 ± 1.47	6.50 ± 2.25	0.079

*: Mean ± SD; **: *n* (%); PD: probing depth; CAL: clinical attachment level.

## Data Availability

All data used to support the findings of this study are included in the manuscript.

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
