# Peer review of "qPCR Detection and Quantification of Aggregatibacter actinomycetemcomitans and Other Periodontal Pathogens in Saliva and Gingival Crevicular Fluid among Periodontitis Patients"

_pathogens, 2023, doi:10.3390/pathogens12010076_

Round 1

Reviewer 1 Report

Dear Authors,

The original manuscript entitled “qPCR detection and quantification of Aggregatibacter actinomycetemcomitans and other periodontal pathogens in saliva and gingival crevicular fluid among periodontitis patients” is appropriately structured, developed and written by Reddahi et al. in suitable English with a clear structure. They investigated the presence of specific periopathogens and studied whether there is a correlation between the results of different bacterial species in the whole saliva and pooled subgingival plaque samples from individuals with periodontitis and periodontally healthy subjects. This study is interesting; however, there are some major and minor concerns which must be addressed.

Main concerns:

-        There is not any reference for the sampling procedure.

-        16srDNA PCR products are suggested to be sequenced to confirm the bacterial species. Please explain why it has not been implemented or do the sequencing.

-        Please add the qPCR melting curves and standard curves into the manuscript as the main or supplementary data.

Minor points:

-        Some bacterial names are not in italics form throughout the manuscript. Please check all of them and revise them throughout the manuscript.

-        Between the point at the end of the sentences and the first word of the next sentence must be spaced. Please check them throughout the manuscript and revise them.

-        p-value should be written in italics form. Please revise them throughout the manuscript.

-        There are some spelling and grammar errors throughout the manuscript. Please revise the manuscript thoroughly by a native English writer. 

Author Response

Answers to Reviewer 1

The original manuscript entitled “qPCR detection and quantification of Aggregatibacter actinomycetemcomitans and other periodontal pathogens in saliva and gingival crevicular fluid among periodontitis patients” is appropriately structured, developed and written by Reddahi et al. in suitable English with a clear structure. They investigated the presence of specific periopathogens and studied whether there is a correlation between the results of different bacterial species in the whole saliva and pooled subgingival plaque samples from individuals with periodontitis and periodontally healthy subjects. This study is interesting; however, there are some major and minor concerns which must be addressed.

Main concerns:

-        There is not any reference for the sampling procedure.

Please would you refer to ‘saliva and subgingival plaque sampling’ part; sampling protocol has been more explained in page 3/ line 106 and references had been added too.

-        16srDNA PCR products are suggested to be sequenced to confirm the bacterial species. Please explain why it has not been implemented or do the sequencing.

In this study, we used specific primers and probes for each  the targeted bacteria, namely P.gingivalis, clone JP2 and non-JP2 of A. actinomycetemcomitans

-       Please add the qPCR melting curves and standard curves into the manuscript as the main or supplementary data.

Please find enclosed supplementary Data Sheet S1 with qPCR standard curves added.

Minor points:

-        Some bacterial names are not in italics form throughout the manuscript. Please check all of them and revise them throughout the manuscript.

Thank you for your remark, italics form of bacterial names had been revised.

-        Between the point at the end of the sentences and the first word of the next sentence must be spaced. Please check them throughout the manuscript and revise them.

Space between the point at the end of the sentences and the first word of the next sentence has been revised.

-        p-value should be written in italics form. Please revise them throughout the manuscript.

 p-value has been corrected in italics form throughout the manuscript. Thank you for your remark.

-        There are some spelling and grammar errors throughout the manuscript. Please revise the manuscript thoroughly by a native English writer. 

Thank you for your comments, errors have been corrected throughout the manuscript.

Reviewer 2 Report

Dear author,

This is a well conducted study but the manuscript needs some corrections.

1) Multiple places space between words missing.

2) Some more relevant papers related to Aa and red complex bacteria, specifically related to P.gingivalis can be added.

3) Statistical analysis does not mention specific test applied to confirm co-relation when the paper speaks of co-relation on several aspects. Was log regression analysis done for clinical parameters or to check the correlation between Aa and Pg?

4) Citation for primers used is missing.

5) Universal 16s Primers are known to amplify all the bacteria and not just the periodontal pathogens and difference in the quantity of the bacteria between health and disease exist is long been known. Can you explain the novelty or what was exactly intended, readers may find it difficult to understand.

6) Line 133-134 should have been mentioned after line 126, if not those lines just add to the confusion.

7) Which indices were used for PI and BOP? Please mention it specifically and cite them.

8) Including / Excluding Criteria: Can be changed to Inclusion / Exclusion

9) Was the inclusion criteria only include Non-smokers or no tobacco users?

10) Subgingival plaque sampling has not been described. GCF sampling was done with only one paper point per tooth? Citation for the same missing. GCF sample extraction for further downstream processing is missing. Line 110, did it mean healthy sites in periodontitis patient?

11) Was the patient grouping based on the current classification, mention of which is missing.

12) Citation : 47 and 50 is repeated.

Author Response

qPCR detection and quantification of Aggregatibacteractinomycetemcomitans and other periodontal pathogens in saliva and gingival crevicular fluid among periodontitis patients

Answers to Reviewer 2

This is a well conducted study but the manuscript needs some corrections.

1) Multiple places space between words missing.

Spaces between words has been corrected.

2) Some more relevant papers related to Aa and red complex bacteria, specifically related to P.gingivalis can be added.

As proposed, some more recent references related to Aa and Pg were added:

  • Charalampakis G, Dahlén G, Carlén A, Leonhardt A. Bacterial markers vs. clinical markers to predict progression of chronic periodontitis: a 2-yr prospective observational study. Eur J Oral Sci. 2013 Oct;121(5):394-402. doi: 10.1111/eos.12080. PMID: 24028586. 5.
  • Fine DH, Markowitz K, Furgang D, Fairlie K, Ferrandiz J, Nasri C, McKiernan M, Gunsolley J. Aggregatibacter actinomycetemcomitans and its relationship to initiation of localized aggressive periodontitis: longitudinal cohort study of initially healthy adolescents. J Clin Microbiol. 2007 Dec;45(12):3859-69. doi: 10.1128/JCM.00653-07. Epub 2007 Oct 17. PMID: 17942658; PMCID: PMC2168549. 6.
  • Haubek D, Ennibi OK, Poulsen K, Vaeth M, Poulsen S, Kilian M. Risk of aggressive periodontitis in adolescent carriers of the JP2 clone of Aggregatibacter (Actinobacillus) actinomycetemcomitans in Morocco: a prospective longitudinal cohort study. Lancet. 2008 Jan 19;371(9608):237-42. doi: 10.1016/S0140-6736(08)60135-X. PMID: 18207019. 7.
  • Fine DH, Markowitz K, Fairlie K, Tischio-Bereski D, Ferrendiz J, Furgang D, Paster BJ, Dewhirst FE. A consortium of Aggregatibacter actinomycetemcomitans, Streptococcus parasanguinis, and Filifactor alocis is present in sites prior to bone loss in a longitudinal study of localized aggressive periodontitis. J Clin Microbiol. 2013 Sep;51(9):2850-61. doi: 10.1128/JCM.00729-13. Epub 2013 Jun 19. PMID: 23784124; PMCID: PMC3754677.
  • Belibasakis GN, Maula T, Bao K, Lindholm M, Bostanci N, Oscarsson J, Ihalin R, Johansson A.: Virulence and Pathogenicity Properties of Aggregatibacter actinomycetemcomitans. Pathogens. 2019 6; 8 (4): 222. doi: 10.3390/pathogens8040222. PMID: 31698835

3) Statistical analysis does not mention specific test applied to confirm co-relation when the paper speaks of co-relation on several aspects.

For quantitative variables with a Gaussian distribution, we used Pearson's correlation test, and for variables with a non-Gaussian distribution, we used Spearman's correlation.

Was log regression analysis done for clinical parameters or to check the correlation between Aa and Pg?

Since it wasn't our goal, the regression for the clinical parameters wasn't performed.

4) Citation for primers used is missing.

Citations for primers are now added both in the text and table 1. Please see page 3 line 140-141.

The selection of primers was based on previous data sources with information on species-specific oligonucleotides of the 16S ribosomal RNA in each strain of this study [15-16].

Please would you refer to Table 1 in page 4, where references had been mentioned too.

5) Universal 16s Primers are known to amplify all the bacteria and not just the periodontal pathogens and difference in the quantity of the bacteria between health and disease exist is long been known. Can you explain the novelty or what was exactly intended, readers may find it difficult to understand.

The Universal 16s probe used in our study, has the capacity to detect 14 oral bacterial species (please refer to line 120/ page 3) tested previously by Kirakodu et al [15]. As we did not comprehensively assess all periopathogenic bacteria, because of lack in means, we opted to present results of universal probe in order to show whether there is a differentiation in the overall oral bacteria between non-periodontitis and periodontitis patients.

The majority of investigations have examined several bacterial species using either plaque or saliva samples. Our major goal was to compare the concentrations of the investigated potential bacteria in salivary and subgingival plaque samples from the same individuals. Since saliva is simpler to collect, this information may be useful in future research to confirm saliva as a diagnostic tool. The following sentence was added to explain the aim of this:

Higher levels of the studied periopathogens were found in both saliva and plaque samples from the periodontitis group in comparison to healthy subjects. That, support previous studies on the bacterial etiology of periodontitis. Indeed, it is largely admitted that periodontitis may occur in presence of complex polymicrobial communities’ whish can threaten the periodontium because the grow of the total number of bacteria or because of development of highly pathogenic bacteria. The complex polymicrobial biofilm associated to periodontitis interact with immune response leading to periodontium breakdown.

6) Line 133-134 should have been mentioned after line 126, if not those lines just add to the confusion.

Thank you for your remark. Please refer to line 129-130, page 3; lines have been moved as proposed.

7) Which indices were used for PI and BOP? Please mention it specifically and cite them.

For periodontal diagnostics, plaque index (PI) (O’Leary et al. 1972) and bleeding on probing (BOP) (Ainamo & bay 1975) were assessed… please see, page 2, line 81-82.

  • O’Leary TJ, Drake RB, Naylor JE. The Plaque Control Record. J Periodontol. 1972; 43:38–38.
  • Ainamo J, Bay I. Problems and proposals for recording gingivitis and plaque. Int Dent J. 1975; 25:229–35.

8) Including / Excluding Criteria: Can be changed to Inclusion / Exclusion

Changes were made. Please refer to line 70 and 71, page 2.

9) Was the inclusion criteria only include Non-smokers or no tobacco users?

The inclusion criteria included …. and no tobacco users. Please see line 71 page 2

10) Subgingival plaque sampling has not been described. GCF sampling was done with only one paper point per tooth? Citation for the same missing. GCF sample extraction for further downstream processing is missing. Line 110, did it mean healthy sites in periodontitis patient?

Line 110, did it mean healthy sites in periodontitis patient?

Subgingival plaque sampling has been described as bellow, please see page 3, line 106-112

GCF samples were collected as pool samples. In the periodontitis group, we collected one pooled sample from diseased sites (two deepest pockets exhibiting the highest CAL values) and one pool sample from two healthy sites when possible.  A pool sample was taken from two healthy sites in healthy subjects too.  Sampling sites were isolated from the saliva by a cotton roll and supra-gingival plaque was removed, after that two sterilized autoclaved medium paper point N°40 were inserted consecutively into the chosen sites and left for 30 seconds. Samples were then put in 500µL of phosphate-buffered saline (PBS) solution.

11) Was the patient grouping based on the current classification, mention of which is missing.

Yes, patients were grouping based on the current classification. Please, see page 2 line 89-90.  

12) Citation: 47 and 50 is repeated.

Thank you for your comment, doubled citation has been removed.

Round 2

Reviewer 1 Report

Dear authors,

Thank you very much for your revising.